# Hypoxia, Ion Channels and Glioblastoma Malignancy

**DOI:** 10.3390/biom13121742

**Published:** 2023-12-04

**Authors:** Antonio Michelucci, Luigi Sforna, Fabio Franciolini, Luigi Catacuzzeno

**Affiliations:** Department of Chemistry, Biology and Biotechnology, University of Perugia, 06123 Perugia, Italy; luigi.sforna@unipg.it (L.S.); fabio.franciolini@unipg.it (F.F.)

**Keywords:** BK channel, cell death, glioblastoma, hypoxia, invasion, volume regulation, VRAC

## Abstract

The malignancy of glioblastoma (GBM), the most aggressive type of human brain tumor, strongly correlates with the presence of hypoxic areas within the tumor mass. Oxygen levels have been shown to control several critical aspects of tumor aggressiveness, such as migration/invasion and cell death resistance, but the underlying mechanisms are still unclear. GBM cells express abundant K^+^ and Cl^−^ channels, whose activity supports cell volume and membrane potential changes, critical for cell proliferation, migration and death. Volume-regulated anion channels (VRAC), which mediate the swelling-activated Cl^−^ current, and the large-conductance Ca^2+^-activated K^+^ channels (BK) are both functionally upregulated in GBM cells, where they control different aspects underlying GBM malignancy/aggressiveness. The functional expression/activity of both VRAC and BK channels are under the control of the oxygen levels, and these regulations are involved in the hypoxia-induced GBM cell aggressiveness. The present review will provide a comprehensive overview of the literature supporting the role of these two channels in the hypoxia-mediated GBM malignancy, suggesting them as potential therapeutic targets in the treatment of GBM.

## 1. Glioblastoma

Glioblastoma (GBM) is the most common and aggressive type of brain tumor. It has been classified by the World Health Organization (WHO) as a grade IV astrocytoma among the four grades (I–IV) of greater progressive malignancy, defined according to clinical and histopathological criteria, such as excessive hypercellularity, endothelial cell hypertrophy, microvascular hyperplasia and necrotic areas. GBM is not curable and patients’ median survival, applying the best standard care possible, is around 15 months from diagnosis [1,2]. The current standard treatment includes surgical tumor resection based on pre-operative magnetic resonance images (if available), followed by radiotherapy and chemotherapy (with temozolomide) [3,4,5]. Full surgical resection of the tumor, on which most hopes for solid tumors are focused, is never feasible or complete for GBM due to its high invasiveness. As result, although surgery can significantly increase patient survival, it is never resolving and eventually tumor progression continues and patient death occurs. The high migratory and invasive potential, which is responsible for the widespread infiltration of tumor cells into the healthy brain parenchyma, and the formation of new foci are therefore the main problems in GBM tumors.

GBM cells invade following different routes, mainly through the brain parenchyma, white matter tracts and blood vessels [6]. Using the magnetic resonance imaging of GBM patients, Esmaeili and coworkers found a preferential direction of GBM cell migration along the white matter fibers [7], confirming earlier studies from Sontheimer’s laboratory [8]. A later study, carried out by combining organotypic brain slices and live imaging to trace the dynamics of GBM cells’ invasion, found that GBM cells migrate preferentially along blood vessels [9]. In any case, cell invasion outside the central mass is the major feature of GBM and the reason for the limited beneficial effects of tumor surgery and the fatal outcome of the disease.

## 2. Hypoxia and GBM Aggressiveness

Extensive tissue hypoxia is one of the main features of GBM and a major determinant of the tumor’s aggressive phenotype. Hypoxia in GBM is the result of the rapid proliferation and expansion of the tumor mass, which the vascular system, although strongly stimulated to grow, is unable to reach, making the tumor microenvironment heavily hypoxic [10]. The hypoxic condition stimulates a variety of biological responses to counteract this adverse situation, starting with the stabilization of the hypoxia-inducible transcription factor 1 (HIF-1), which activates the expression of several hypoxia response genes [11]. These primarily include vascular endothelial growth factor, which stimulates the proliferation of endothelial cells and the formation of new blood vessels to supply the growing tumor mass with adequate blood and oxygen [12]. However, this neovascularization develops aberrantly, producing large, tortuous and malfunctioning new vessels that are highly prone to vascular thrombosis and vessel occlusion, which in turn leads to severe hypoxia and acidosis [13]. These conditions, which the tumor cells cannot tolerate for long, eventually lead to massive cell death and the formation of the necrotic areas (foci) observed histologically, which are one of the most distinctive features of GBM (Figure 1) [14].

Surviving tumor cells escape the heavily hypoxic regions, heading for more oxygen-richer areas near functioning blood vessels, where they form dense cell fronts called pseudopalisades, assemblages of elongated tumor cells stacked in rows on the periphery of the necrotic regions formed around the occluded vessels [15]. This view is based on the observation that pseudopalisading cells are much less proliferative and more prone to apoptosis than adjacent tumor cells, ruling out the possibility that these dense cell fronts do not result from increased clonal expansion or resistance to apoptosis [16,17,18]. It is this cell front that mainly spreads into and invades healthy brain tissue, making the surgical resection of the tumor problematic and ultimately being the principal cause of death in GBM patients [19].

**Figure 1 biomolecules-13-01742-f001:**
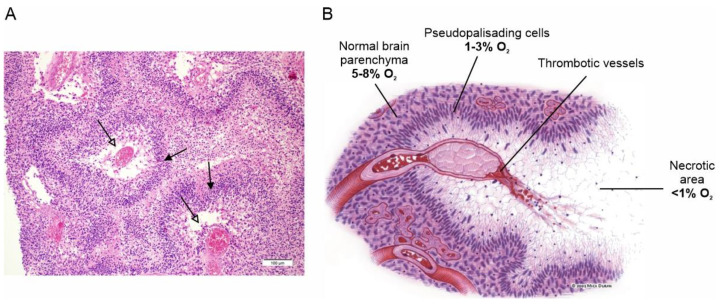
Necrotic areas and pseudopalisades, the distinctive histopathological features of GBM. (**A**) Classical histologic findings of GBM showing prominent pseudopalisading necrosis (empty arrows). Indented area of necrosis with pseudopalisade formation characterized by accumulation of escaping tumor cells (filled arrows). Adapted with permission from Ref. [20]. © 2015 Lim et al. (**B**) Schematic representation of pseudopalisade formation in GBM. Vaso-occlusion/collapse and intravascular thrombosis lead to tissue hypoxia in the perivascular region. Hypoxic tumor cells then migrate away along cellular processes, leaving initially a fibrillar center. Tumor cells that do not migrate become hypoxic and undergo apoptosis or necrosis, eventually leaving a central necrotic zone. Adapted with permission from Ref. [21]. © 2004 USCAP, Inc.

Besides increasing their migratory ability to survive the adverse hypoxic conditions, GBM cells implement several other changes, such as (i) switching from aerobic to anaerobic metabolism [22,23]; (ii) promoting the selection and maintenance of the more aggressive glioblastoma stem cells [24,25,26]; (iii) upregulating a second form of HIF, HIF-2, which increases erythropoietin production in the kidney and liver, resulting in increased hemoglobin synthesis [27] and endowing GBM cells with properties that help them to evade apoptotic death [28].

Among the many effects induced by hypoxia, two—cell migration/invasion and resistance to cell death—are strongly correlated with cell volume regulation [29,30,31,32,33], a complex process under the strict control of transporters and ion channels localized on the plasma membrane [34]. These observations would suggest the possible involvement of volume-regulated ion channels in the hypoxia-induced aggressiveness of tumor cells.

## 3. Volume-Regulated Ion Channels in GBM

The volume-regulated anion channel (VRAC) is widely expressed in vertebrate cells, including GBM tumors, where it mediates the swelling-activated Cl^−^ current (I_Cl,swell_) required for cell volume regulation [35,36,37,38,39]. Electrophysiological studies characterized the VRAC currents more than 30 years ago [40,41], but the molecular identity of the channel has been a source of uncertainty and controversy for decades [42]. In 2014, two laboratories, independently, reported that VRAC is a heteromeric channel encoded by different members of the *Lrrc8* gene family [43,44].

As the name implies, VRAC is sensitive to volume changes, so routinely VRAC is activated by an increase in cell volume, generally obtained by exposing the cell to hypotonic solutions. Other distinguishing features of VRAC are its significant permeability to many anionic species (SCN^−^, I^−^, NO_3_^−^, Br^−^, Cl^−^, HCO_3_^−^, F^−^, glycine) and the generation of outward rectifying currents. VRAC has also been found to pass large molecules, such as taurine and glutamate, and several anticancer drugs, such as cisplatin and carboplatin, which makes this channel the main one responsible for cancer drug resistance [45,46].

In GBM cells, when electrophysiological recordings are performed in the presence of Ca^2+^-activated K^+^ channel inhibitors, the application of a hypotonic solution leads to the activation of a current with a reversal potential very close to the Cl^−^ equilibrium and completely blocked by VRAC-mediated I_Cl,swell_ current antagonists such as NPPB, DIDS and DCPIB [39,47,48]. The slow inactivation of the hypotonic-induced VRAC-mediated I_Cl,swell_ at high voltages is another typical feature of the current [49]. We also investigated in detail the signal transduction pathway that leads to swelling-induced VRAC activation. Our data suggest that cell swelling activates a volume-sensitive PLC and increases DAG levels. DAG is then phosphorylated by DGK to phosphatidic acid, which is known to activate the Rac1 G protein. Activated Rac1 will then induce the polymerization of cortical actin, which is the final step needed for the activation of the VRAC channel [39]. Notably, this transduction pathway is very different from that proposed for astrocytes, which represent the normal cells from which GBM cells originate [50,51,52].

In addition to responding to osmotic shock, VRAC is modulated by severe hypoxia or elevated serum content, both of which are typical of the tumor environment and known to favor glioma malignancy [47]. Other intracellular signals, such as ATP, cytoplasmic Ca^2+^, reactive oxygen species, G-proteins and phosphorylation, have been reported to activate VRAC. None, however, has shown its efficacy across different cell types to gain general significance.

Regarding the role of VRAC in GBM malignancy, Sontheimer and coworkers suggested that the infiltration of GBM cells into the healthy brain parenchyma requires major changes in cell shape and volume and that Cl^–^ currents are crucial in this process [53,54]. Later studies added new roles for VRAC in GBM malignancy, such as in cell proliferation, escape from cell death and multidrug resistance [48,49,55,56,57]. However, these roles must be considered with caution because, prior to the molecular identification of VRAC in 2014, recorded currents were assigned to VRAC mainly based on studies using pharmacological blockers that were not very specific to VRAC, such as DIDS and NPPB [58,59,60]. Unfortunately, these uncertainties precluded a thorough investigation of its pathophysiological roles in GBM and the underlying mechanisms for many years. Indeed, some discrepancies regarding the role of VRAC in cell proliferation and migration have been reported in the literature. In a recent study, Liu and Stauber demonstrated that the genetic ablation of the LRRC8A subunit, which is essential for the establishment of a functional VRAC, affected neither proliferation nor migration in two well-established GBM cell lines, concluding that VRAC is not essential for either process [61]. By contrast, though, Serpe and colleagues showed that the reduction of both the LRRC8A and LRRC8C subunits, induced by treating GL261 murine GBM cells with astrocyte-derived exosomes, significantly hampered both migration and invasion [62]. The reasons for the discrepancies have not yet been fully clarified.

Another important process in which VRAC plays a central role is observed when GBM cells are placed in a hypotonic solution. After the immediate swelling of the cell due to water rebalancing caused by the osmotic gradient across the membrane, the cell slowly tries to re-establish its original cell volume, a process called regulatory volume decrease (RVD). The mechanism, now clearly established, is that VRAC channels, activated by membrane stretching and the PLC-DAG-phospatidic acid pathway, together with activated K^+^ channels, extrude KCl salt. The osmotic imbalance created is followed by the efflux of water, which restores the volume of the swollen cell to its original size. This process is often used experimentally to verify the involvement of specific ion channels in the regulation of the cell volume [63]. Recently, we confirmed the role of VRAC channels in the RVD process in a murine GBM cell line using DCPIB, a selective VRAC inhibitor [62].

### The Ca^2+^-Activated K^+^ Channels

Another extensively studied channel family, firmly associated with cell volume control and GBM migration and invasion, is that of the Ca^2+^-activated K^+^ (K_Ca_) channels, which includes the large-conductance (~250 pS) (K_Ca_1.1, or BK), the intermediate (20–80 pS; K_Ca_3.1 or IK) and the small-conductance (12–20 pS; K_Ca_2.1–2.3 or SK 1–3) channels [64]. A common and distinctive property of this channel family is their activation by intracellular Ca^2+^, although with mechanisms and sensitivities that are considerably different. All types of K_Ca_ channels are expressed in GBM cells; however, only BK and IK channels have emerged as the K_Ca_ channels certainly associated with GBM malignancy, being involved in cell migration and invasion, cell proliferation and escape from cell death [53,65,66,67].

The BK channel belongs to the voltage-gated K^+^ channel family, having the typical structure of four α subunits forming the channel, each composed of six transmembrane segments (S1–S6) plus an extra one, the S0, with charged S4 segments that confer upon the channel the sensitivity to voltage [68,69]. Its Ca^2+^ sensitivity is instead imparted by intrinsic Ca^2+^ binding sites (collectively called the Ca^2+^ bowl) in their carboxyl terminal tail [70]. The Ca^2+^ sensitivity of the channel is modest, with a Kd ~3 µM at 0 mV [71]. As result, co-localization with Ca^2+^-permeable channels is needed to provide local Ca^2+^ sufficient to activate the channels [72]. Given the double sensitivity to both Ca^2+^ and voltage, the BK channel is activated cooperatively by a cytoplasmic Ca^2+^ increase and membrane depolarization [73,74,75,76]. Given their very high conductance, the activation of BK channels leads to the massive efflux of K^+^ ions, which hyperpolarizes the cell membrane but may also serve other purposes.

By combining biochemical and pharmacological approaches, Sontheimer and coworkers found that BK channels were highly overexpressed in several GBM cell lines (D54-MG, U251-MG, STTG-1), as well as in primary cultures of glioma cells established from patient biopsies [77,78]. Moreover, when analyzing biopsies from patients with WHO grade I to IV gliomas, it was found that BK channel expression correlated with the tumor grade [79]. Several other studies have shown that BK channels are overexpressed in malignant glioma cells [65,80,81,82].

BK channels were suggested very early and repeatedly to be involved in GBM migration and invasion. Evidence was obtained by testing the selective BK channel blocker iberiotoxin on GBM transwell migration [53] or on migration induced by different agents that increase cytoplasmic Ca^2+^, such as acetylcholine [83], phloretin [84] and menthol [85]. In all these cases, the pharmacological inhibition of BK channels affected cell migration. Ionizing radiation (IR) has also been shown to enhance Ca^2+^ signaling and stimulate cell migration of GBM cells [86,87,88]. Huber and coworkers first showed, in an in vitro study, that IR-induced BK channel activation is a key event in IR-induced GBM migration, and that the BK channel blocker paxilline suppresses it [86]. These results were later confirmed by the same laboratory in vivo [87]. Similar results were reported for another K_Ca_ channel by Limatola and coworkers. They demonstrated that high-dose radiation of patient-derived GBM cells increased the functional expression of IK channels, which correlated with the pro-invasive phenotype of tumor cells induced by IR [88].

The IK channel is the other member of the K_Ca_ channel family that has been most investigated in the context of GBM invasion [66,67,89]. The expression of IK currents and IK channel transcripts on established GBM cell lines (GL15 and U251) was first reported by our laboratory [90]. IK channel transcripts (but not the current) were also found by Sontheimer’s group in the U251 and D54-MG GBM cell lines [78]. The IK channel was later found to be overexpressed in ~30% of glioma patient biopsies, where it correlated with poor patient survival [67].

We also demonstrated the involvement of IK channels in cell migration in human GBM cell lines and in freshly dissociated tissues from GBM patients. Evidence for major IK channel involvement in human GBM cell migration and invasion was later provided in vivo in severe combined immunodeficiency (SCID) mice xenografted with GBM GL15 cells [91]. Namely, IK channel inhibition by shRNA or specific blocker TRAM-34 markedly reduced the tumor-infiltrated area (especially along the white matter tracts) and the antero-rostral spreading of GBM cells in the brain parenchyma [91]. These results were later replicated by Sontheimer’s laboratory both in vitro and ex vivo, using patient-derived glioma tissues xenografted in the flanks of nude mice, as well as U251 GBM cell lines [67]. IK channels also promote GBM cell migration by inducing or modulating cytoplasmic Ca^2+^ signals in the form of Ca^2+^ oscillations [92,93,94]. Altogether, these results confirm the crucial role of IK channels in brain tumor cell migration and invasion.

We recently reported that both BK and IK currents were significantly activated by exposure to a 30% hypotonic solution [95]. Since both K^+^ channels are activated by Ca^2+^ and since cell swelling increases Ca^2+^ in many cell types, we tested whether Ca^2+^ influx was needed for the hypotonic-induced BK and IK channel activation. We found that the activation of both channels was prevented by incubating the cells in extracellular solutions with nominally zero Ca^2+^. In addition, the hypotonic-induced activation of both BK and IK currents could also be prevented by the unspecific inhibitor of mechanosensitive channels, gadolinium (Gd^3+^). This suggests that, following cell swelling, Ca^2+^ influx through mechanosensitive channels mediates the activation of both K^+^ channels [95]. Ca^2+^-permeable mechanosensitive channels (MSCs) expressed in eukaryotic cells were found to have a single-channel conductance of 20–40 pS and to be inhibited by Gd^3+^, by the spider toxin GsMTx4 and by streptomycin [96,97,98]. In 2010, two channels, Piezo1 and Piezo2, were identified as the long-sought structural counterparts of the non-selective cationic MSC current found in many cells [99]. Indeed, Piezo1 expression in heterologous cell types was found to replicate the gating properties of the endogenous, ubiquitous non-selective cation channels in patches, producing elementary currents of 20–50 pS with rapid voltage-dependent inactivation, and inhibited by GsMTx4 [100,101]. The Piezo1 channel is a very large protein of over 2500 amino acids and was initially thought to be formed by tetramers [99]. However, the recently obtained cryo-EM structure of Piezo1 clearly shows that the channel is a trimer [102]. Structure–function studies suggest that Piezo1 opening occurs upon the application of a wide range of mechanical forces on the plasma membrane, including membrane stretching [99,103]. However, Piezo1 has highly selective activators, such as yoda1 and jedi2, that can open the channel in the absence of mechanical stimuli [104,105].

Notably, it has been recently found that Piezo1 channels are highly expressed in gliomas and their expression correlates with the tumor grade and malignancy [106,107,108]. In addition, a recent study reported that the yoda1-mediated activation of Piezo1 in isotonic conditions reduced GBM cells’ resistance to TNF-α-related apoptosis-inducing ligand (TRAIL), significantly increasing TRAIL-mediated apoptosis in two different GBM cell lines [109].

In accordance with these results, in GBM cells we found that IK and BK channels could also be activated under isotonic conditions by the selective Piezo1 agonist yoda1. We also reported that, in the simultaneous presence of DCPIB, TRAM-34 and paxilline, both yoda1 and hypotonic solutions activated a strongly outward-rectifying, cation-unselective current resembling the current observed by others after the exogenous expression of Piezo1 [95,110].

In addition to responding to the increase in cell volume, IK and BK channels also participate in the regulation of the GBM cell volume, as assessed using the RVD paradigm. More specifically, in U87-MG GBM cells, the exposure of cells to a 30% hypotonic solution induced rapid swelling followed by the slow return of the original cell volume. Notably, both paxilline and TRAM-34, selective inhibitors of the BK and IK channels, respectively, significantly hamper the RVD process (Figure 2A,B) [95]. All these findings are synthetized in the mechanistic model of RVD shown in Figure 2C, illustrating how the selective blockade of each of the ion channels mentioned above (i.e., VRAC, IK, and BK) is able to significantly inhibit the RVD process (Figure 2A,B). The hypotonic-induced cell swelling causes an increase in the plasma membrane tension/stretching, which represents the stimulus for the opening of mechanosensitive Ca^2+^-permeable channels, such as Piezo1. The resulting increment in the intracellular Ca^2+^ concentration, due to Piezo1-mediated Ca^2+^ entry, triggers the activation of both K_Ca_ channels. VRAC channels, which in GBM cells respond to the hypotonic shock in a Ca^2+^-independent manner [39], open instead via the activation of the PLC/DAG/Rac1 pathway. However, we cannot rule out that the opening of Piezo1 may also modulate the VRAC-mediated I_Cl,swell_, as we reported in HEK293 cells [111]. In any case, the net efflux of KCl creates the osmotic gradient necessary for the loss of water and the consequent restoration of the original cell volume (Figure 2C).

## 4. Ion Channels in GBM Cell Migration and Death

The basic mechanism underlying the migration of GBM cells, indeed of cells in general, can be described as the cyclic succession of two distinct processes: the protrusion of the cell front, due to actin polymerization with the formation of pseudopods, and the retraction of the rear cell body, due to forces produced by actomyosin contraction. Both processes, to develop properly, must be accompanied by the local remodeling of the cell volume and shape [112,113,114], a notion confirmed by live imaging studies in glioma cells [8]. These major remodeling of the cell shape and volume are made possible by the complex interplay of ion channels and transporters outlined below [49,67,114,115,116].

At the leading edge of GBM cells, the activation of the Na^+^/K^+^/2Cl^−^ cotransporter (NKCC1), which brings one Na^+^, one K^+^ and two Cl^–^ ions inside the cell, accompanied by the iso-osmotic influx of water, leads to an increase in cell volume [117]. For the cell to move forward, the protrusion of the cell front must be followed by the retraction of the rear end, which, as shown in Figure 3, results in a significant decrease in cell volume by the extrusion of ions (i.e., K^+^ and Cl^−^) and osmotic water [112,113,114]. More precisely, the retraction of the trailing edge begins with the activation of MSCs and VRAC, which occurs in response to the stretching of the membrane caused by the NKCC1-dependent increase in cell volume. The activation of the MSCs and the consequent Ca^2+^ influx activate the K_Ca_ channels and the efflux of K^+^ ions that, in combination with the efflux of Cl^−^ through the VRAC and osmotic water, results in the reduction of cell volume required for the retraction of the rear end (Figure 3).

These hydrodynamic events, involving the concerted transmembrane flow of ions and osmotic water and consequent changes in shape and volume, help GBM cells to invade the brain parenchyma. It is important to emphasize, however, that they only have a permissive or facilitatory role in the process, with no implications for tumor initiation [112].

Volume changes induced by ion channels are also crucial for cell death. In GBM, K_Ca_ channels have been implicated in the so-called apoptotic volume decrease (AVD), a mechanism whereby the cell reduces its volume before apoptosis (Figure 4, left branch). Using a grade IV human GBM cell line, Sontheimer and coworkers examined the contribution of K_Ca_ channels to AVD after the addition of either staurosporine or TRAIL to activate the intrinsic or extrinsic pathway of apoptosis, respectively [118]. The use of specific K_Ca_ channel inhibitors revealed that staurosporine-induced AVD was dependent on K^+^ efflux through IK channels, while TRAIL-induced AVD was mediated by BK channels. Conversely, an increase in cell volume, called necrotic volume increase (NVI), is observed during necrosis (Figure 4, right branch). This is mainly caused by the influx of NaCl, due to the reduced energy supply and activity of the Na^+^/K^+^ pump, followed by the osmotically driven entry of water. This second type of death is particularly interesting in this context, since GBM cells need to develop strategies to resist hypoxic-induced necrosis.

## 5. Hypoxic Modulation of Volume-Regulated Ion Channels in GBM

We have seen that the activity of the volume-regulated channels expressed in GBM cells can control at least two of the functional processes also regulated by hypoxia, namely cell migration/invasion and cell death. It is therefore possible that hypoxia regulates these processes by modulating the activity of the same ion channels. Below, we show evidence for such hypoxia-induced modulation of GBM ion channels.

### 5.1. VRAC Modulation by Hypoxia

It has long been known that hypoxia induces cell swelling in various cellular models [120,121,122], which led us to postulate that VRAC could be activated by hypoxic stimuli. Electrophysiological experiments in a voltage-clamp configuration, designed to specifically study VRAC activation, show that the application of hypoxic solutions (pO_2_ = 1–3%) activates a current with biophysical (i.e., mild outward rectification and time- and voltage-dependent inactivation) and pharmacological (i.e., DCPIB sensitivity) properties congruent with VRAC. The demonstration that the hypoxia-induced increase in cell volume is a crucial step in VRAC activation is further supported by the observation that the application of a 30% hypertonic solution, which induces cell shrinkage, dramatically inhibits the hypoxia-activated VRAC current [47].

Consistent with the fact that GBM cells in vivo experience persistent hypoxia, their chronic (24 h) exposure to a hypoxic condition, known to induce long-term HIF-1-dependent responses to hypoxia, strongly decreases the VRAC current elicited by acute application of the hypoxic solution, as well as the fraction of hypoxic-responsive cells [47]. The reduction in the VRAC current is arguably the consequence of the decreased expression of VRAC channels, as suggested by the even greater reduction in the current when probed with an extracellular hypotonic solution. Interestingly, the downregulation of VRAC channel expression induced by chronic hypoxia significantly affects the ability of GBM cells to recover the original cell volume, further indicating the central role of VRAC in the RVD process [47].

### 5.2. Hypoxia-Induced VRAC-Mediated Volume Decrease Counteracts Necrotic Cell Death

Necrotic or unprogrammed cell death is caused by excessive chemical or physical stress, including severe oxygen deficiency, which leads to an uncontrolled increase in volume and eventually to the rupture of the cell membrane [123,124]. The activation of VRAC in this context can therefore counteract or limit the cell volume increase, thereby promoting the survival of GBM cells that experience hypoxic conditions immediately after the formation of the thrombotic vessel. This function would be lost or attenuated upon chronic exposure to hypoxic conditions, due to the long-term downregulation of VRAC. We can therefore hypothesize that VRAC activation represents a pro-survival factor for GBM cells acutely exposed to hypoxia, increasing their likelihood of remaining alive until they acquire the promigratory phenotype that will help them to escape towards more oxygenated regions of the brain.

We tested this rationale, illustrated in Figure 5A, in a study published in 2017, in which we showed that the hypoxic activation of VRAC currents opposes both hypoxia-induced cell swelling (assessed by the RVD protocol) and necrotic cell death (by the lactate dehydrogenase assay, LDH) [47]. As reported in Figure 5B, hypoxia induces initial transient cell swelling, followed by the essentially complete recovery of the original cell volume (in the continuous presence of hypoxic solution). Repeating the experiment with the only difference of adding the VRAC inhibitor DIDS to the hypoxic solution, to appreciate the role of VRAC in the process, the rate of cell volume recovery was found to be much slower and never complete (Figure 5B). Interestingly, the inhibition of VRAC upon chronic hypoxia significantly increases the number of necrotic cells (Figure 5C), consistent with the notion that increased VRAC activity makes tumor cells more resistant to necrotic death under hypoxic conditions.

### 5.3. Hypoxia Enhances Activation of BK Channel and Promotes Migration of GBM Cells

Another key channel regulated by hypoxia and involved in GBM malignancy is the BK channel [82,125,126]. The acute application of hypoxic solutions to U87-MG cells with varying internal Ca^2+^ concentrations (from nominally zero to 300 nM Ca^2+^) had no effect on the activation of the BK currents, suggesting that acute hypoxia has no effect on the combined voltage/Ca^2+^ gating mechanism of BK channels [125]. Conversely, chronic treatment (24 h) increases significantly the BK current, as assessed by stimulating the cells with depolarizing stimuli (Figure 6A, inset). However, the increased current does not correlate with increased BK channel expression [125] but is rather the result of a marked shift in the half-activation voltage of the hypoxia-conditioned cells towards more hyperpolarized potentials (Figure 6A, main), suggesting that the hypoxic condition significantly affects the gating properties of the BK channel.

Consistent with the role played by BK channels in cell migration and invasion and considering the striking effects of chronic hypoxia on BK currents, the BK current enhancement induced by hypoxia could increase the migration/invasion ability of these cells, in line with its significant role in these processes. We tested this view by using the three-dimensional transwell migration assay. The results obtained suggest that the increased activation of the BK current by chronic hypoxia is instrumental in increasing the invasive activity of GBM cells and consequently the malignancy of the tumor (Figure 6B). These results are fully congruent with our previous wound healing assay, which showed that the hypoxia-induced migration enhancement is significantly reduced in the presence of the BK channel inhibitor paxilline [125]. The results obtained suggest that the increased activation of the BK current by chronic hypoxia is instrumental in increasing the invasive activity of GBM cells and consequently the malignancy of the tumor.

## 6. Conclusions

The data currently available in the literature show the crucial role played by VRAC and BK channels in the formation and organization of the pseudopalisading necrosis and the increased aggressiveness of GBM (Figure 7). According to current views, immediately after vessel occlusion, the resulting hypoxia brings most tumor cells to death, forming the necrotic regions of the tumor mass. Instead, in a certain percentage of cells, VRAC activation is able to oppose the necrotic volume increase and allow survival long enough to acquire a more aggressive and migratory phenotype. Among the features acquired after a sufficiently long time spent in hypoxia is the increased activity of BK channels, which would increase the ability of GBM cells to migrate towards more oxygenated regions and lead to the formation of the characteristic pseudopalisades.

## 7. Future Perspectives

Although the role of IK channels in the invasiveness and migration of tumor cells and in the malignancy of GBM has been largely demonstrated, no data are currently available in the literature on the regulation of this channel by hypoxia. Given the relevant role it plays in GBM malignancy, it is possible that the IK channel, like VRAC and BK channels, is also modulated by the hypoxic insult, possibly through different pathways, and exerts important roles under the hypoxic conditions of GBM. We therefore plan to address these aspects in future studies. Besides the IK channel, another important channel should be investigated for its response to hypoxia, as well as for its role in GBM malignancy: the Piezo1 channel. First, because it is highly expressed in GBM, and its expression correlates with the grade of GBM malignancy. Second, because Piezo1 has been shown to contributes to the regulation of cell volume (and migration) through the modulation of VRAC in HEK293 cells [111], and of both IK and BK channels in U87-MG cells [95].

Finally, our data point to inhibitors of BK, IK and VRAC channels as possible adjuvants in the therapy of this lethal tumor. This prospect of an ion channel-based adjuvant GBM therapy could easily be tested, given the following observations. First, FDA-approved neuroleptics such as haloperidol or chlorpromazine, which inhibit BK channels, are currently undergoing a phase I dose escalation study investigating the repurposing of chlorpromazine adjuvant to radio-chemotherapy with temozolomide for the treatment of newly diagnosed glioblastoma (ClinicalTrials.gov identifier number NCT05190315) [127]. Second, the IK channel inhibitor Senicapoc was well tolerated by patients in clinical trials studying the effect of pharmacological IK channel targeting on sickle cell crisis [128].

## Figures and Tables

**Figure 2 biomolecules-13-01742-f002:**
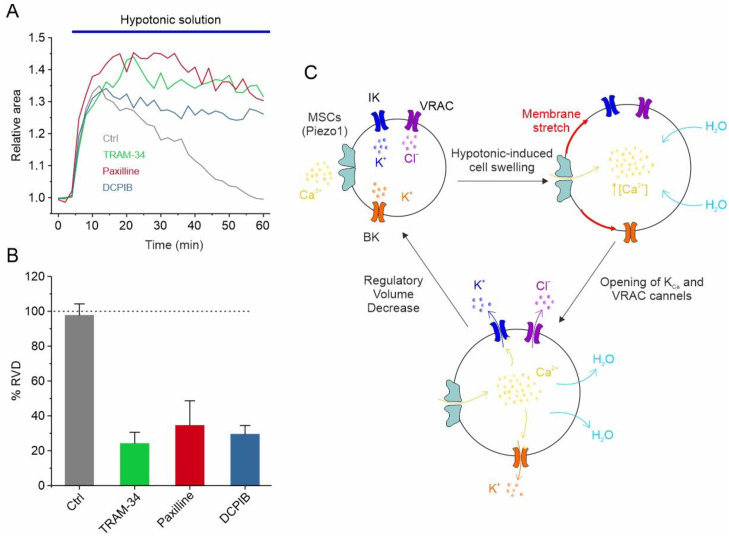
VRAC, IK and BK channels regulate RVD GBM cells. (**A**) Time course of RVD, as evaluated from changes in the relative cell area during application of 30% hypotonic solution (blue bar), in control conditions (grey line) and when either VRAC (blue line), IK (green line) or BK (red line) channel was pharmacologically blocked by 10 μM DCPIB, 3 μM TRAM-34 and 1 μM paxilline, respectively. (**B**) Bar plot showing the %RVD assessed in the different experimental conditions. (**C**) Possible mechanisms underlying the hypotonic-induced cell swelling and the activation of the RVD process. Adapted with permission from Ref. [95]. © 2023 The Authors. Journal of Cellular Physiology published by Wiley Periodicals LLC.

**Figure 3 biomolecules-13-01742-f003:**
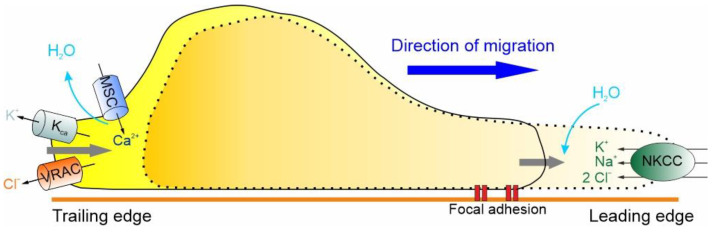
The basic hydrodynamic model of GBM cell migration. GBM cells express ion channels and transporters that underlie shape and volume changes, steps required for cell migration. These include, in the leading edge, NKCC1, and, in the trailing edge, VRAC, K_Ca_ and the MSC channels. The migration cycle includes the uptake of Na^+^, K^+^ and Cl^−^ ions al the leading edge, which attracts osmotic water, resulting in a volume increase and leading edge protrusion. This stretches the membrane and activates both MSC and VRAC at the trailing edge. The resulting Ca^2+^ influx activates the K_Ca_ channels, determining in this way the efflux of K^+^ and Cl^−^ ions, followed by osmotic water, and a decrease in volume at the trailing edge, promoting its retraction.

**Figure 4 biomolecules-13-01742-f004:**
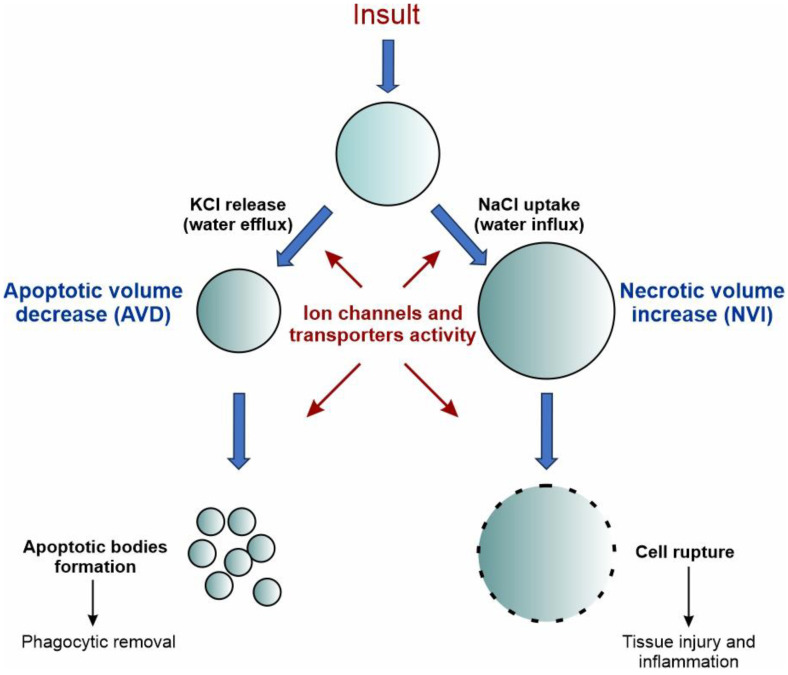
Schematic representation of the role of ion channels in the regulation of cell volume during cell death. Upon persistent death insult, cell shrinkage and cell swelling are hallmarks of the early phases of apoptotic and necrotic cell death, respectively. The early-phase shrinkage of apoptotic cells is termed apoptotic volume decrease (AVD) and is mediated by Cl^−^ and K^+^ channels and the net efflux of KCl, which promotes the osmotic loss of water from the cell. The early-phase swelling of necrotic cells is termed necrotic volume increase (NVI) and is mainly mediated by the transport of Na^+^ and Cl^−^ ions inside the cell, with the resulting influx of water from the extracellular environment. Inspired by [119].

**Figure 5 biomolecules-13-01742-f005:**
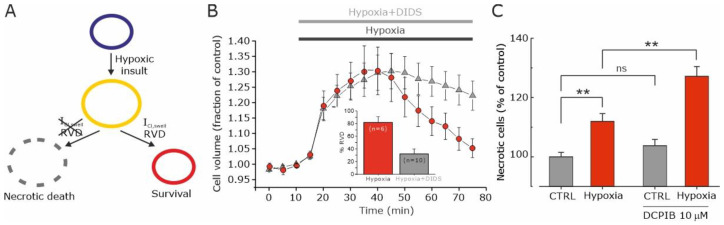
Hypoxia induces VRAC-mediated volume decrease that prevents necrotic cell death. (**A**) Scheme illustrating the postulated role of the hypoxia-activated VRAC current in GBM cell survival in hypoxia-induced necrotic cell death. Following a hypoxic insult, cells undergo an increase in cell volume, which is followed by the activation of the VRAC-mediated I_Cl,swell_ current and the consequent RVD. In the absence of VRAC-mediated I_Cl,swell_ current activation, there is no RVD and cell swelling persists for a longer period, leading to necrotic cell death. (**B**) Time course of the relative cell area in U87-MG cells exposed to a hypoxic solution (red circles) and to a hypoxic solution containing the VRAC channel blocker DIDS (grey triangles). Inset: Bar plot showing the % RVD at 60 min in U87-MG cells exposed to hypoxic solution with and without DIDS. (**C**) Number of necrotic cells (percent of control), measured with the LDH assay, after a hypoxic insult, in absence and presence of the VRAC channel inhibitor DCPIB (10 μM). The hypoxic insult consisted of U87-MG cells’ re-suspension in a medium previously conditioned in a hypoxic chamber. ** *p* < 0.01; ns = not statistically significant. Adapted with permission from Ref. [47]. © 2016 Wiley Periodicals, Inc.

**Figure 6 biomolecules-13-01742-f006:**
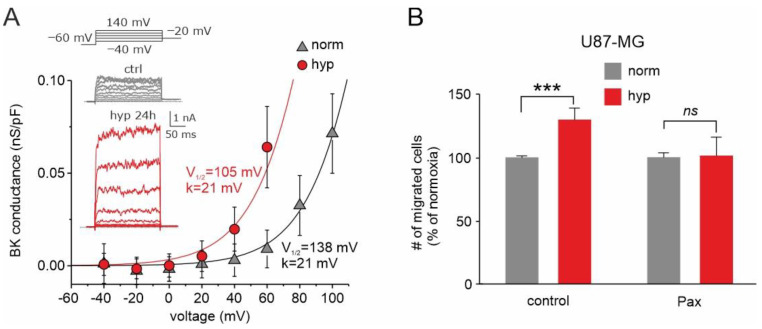
Effects of chronic hypoxia treatment on both BK current and U87-MG cells’ migration. (**A**) Mean BK channel conductance vs. voltage relationships assessed from cells grown either in normoxia or hypoxia. The BK channel conductance was obtained by subtracting the leak currents from the total currents and dividing the resulting pure BK currents for the driving force under our recording conditions. The solid lines represent the best fit of the experimental data with Boltzmann relationships having identical maximal conductance (0.5 nS/pF) and voltage steepness (21 mV), but variable half-activation voltage (105 and 138 mV in hypoxia and normoxia, respectively). Inset: Representative BK currents from control cells, held in normoxia (**upper**, grey traces), and cells held for 24 h in hypoxia (**bottom**, red traces). Stimulation protocol consisted of depolarizing voltage steps from −40 to +140 mV, from a holding potential of −60 mV. (**B**) Transwell migration assay showing the percent of migrated U87-MG cells in normoxia (grey bars) compared to hypoxia (red bars), tested under control conditions and in presence of paxilline. This assay highlights a markedly different effect of hypoxia in the presence and absence of paxilline. *** *p* < 0.001; ns = not statistically significant. Adapted with permission from Ref. [125]. © 2018 Wiley Periodicals, Inc.

**Figure 7 biomolecules-13-01742-f007:**
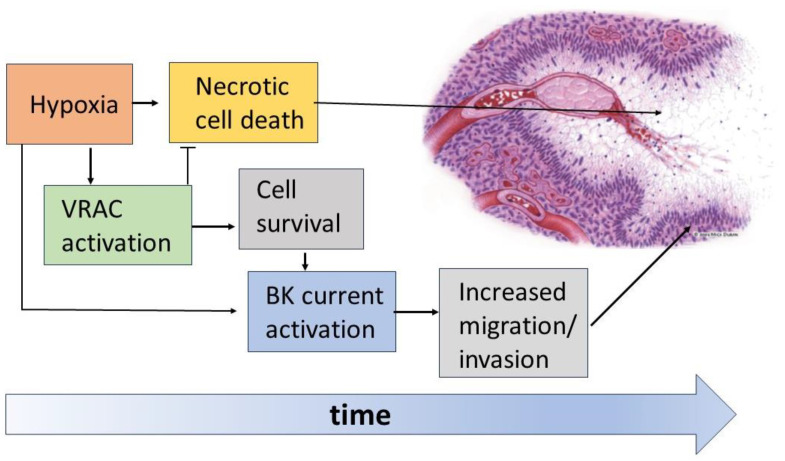
Schematic of the hypoxia-induced events in the organization of pseudopalisades. The hypoxic conditions promote massive cell death with the formation of necrotic areas. However, the hypoxic insult is also the stimulus for the activation of VRAC and BK channels in GBM cells. The enhanced activity of VRAC would help cells to face these challenging conditions, thus rendering GBM cells less prone to necrosis. On the other hand, survived GBM cells use the enhanced activity of BK channels to increase their migratory/invasive potential, thus escaping the hypoxic/necrotic areas.

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
