# Peer review of "Hypoxia, Ion Channels and Glioblastoma Malignancy"

_biomolecules, 2023, doi:10.3390/biom13121742_

Round 1
Reviewer 1 Report
Comments and Suggestions for Authors
The authors present here a review about the involvement of ion channels in the aggressiveness of glioblastoma, especially in hypoxia context. The subject is very interesting and the data presented here are mostly up-to-date. The work could thus be published after minor modifications suggested just above.
The authors have to deeply check the bibliography and clarify some details about channels and hypoxia answer.
Minor comments:
Line 21: please Add VRAC and BK channels in the keywords
Line 37: ref 3 is not supporting the message of the sentence, please check it.
Lines 60-63: is there information about the other members of the HIF family (HIF-2 or HIF-3) ? Please add note about this point
Line 71: ref 13 is not supporting the message of the sentence, please check it.
Figure 1, panel A: it should be interesting to add arrows to accurately describe the different structures presented in the legend.
Line 93: please explain de GSC terms or add it in the abbreviations’ list
Line 97: ref 29 is not supporting the message of the sentence, please check it.
Lines 142-143: the authors discuss the veracity of the studies based on the pharmacology used but the cited molecules are not the same than in the supporting literature that mainly concerned the DCPIB.
Lines 166-169: BKCa is sensitive to voltage but its structure is slightly different by presenting 7 transmembrane segments. In addition, there is more accurate citations to describe it like: Aldrich R, Chandy KG, Grissmer S, Gutman GA, Kaczmarek LK, Wei AD, Wulff H. Calcium- and sodium-activated potassium channels (KCa, KNa) in GtoPdb v.2023.1. IUPHAR/BPS Guide to Pharmacology CITE. 2023; 2023(1). Available from: https://doi.org/10.2218/gtopdb/F69/2023.1. Thus authors have to modify the beginning of this paragraph.
Lines 188-190: authors have to precise that the radiation impacts cell migration by affecting KCa3.1 channel and not BKCa. In addition, the authors present following reference preenting that BKCa is also sensitive to irradiation.
Line 203: an e is lacking at the end of the word where
Line 226: authors have to precise the meaning of MSC
Line 259: there is a typo in independent
Lines 284-305: it should be interesting to discuss briefly the role of the other actors of migrations like enzymes and cell surface receptors. The presentation is good but should be precise by adding other details of the cell migration process.
Line 374: there is a typo in apaper (end of the line)
Lines 405 and 408 and 419: authors used the expression “Data not shonw” but the given information are presented in the previous paper about they discussed.
Figure 6: there is a typo in the y-axis of the panel A (conductance)
Line 424: please, modify the title of the figure because it omits the electrophysiological parts of the results.
Figure 7: authors have to add a scale on time axis because it is difficult to understand if the effect are on short or long period.
Reviewer 2 Report
Comments and Suggestions for Authors
This is a strong manuscript addressing the role of Ca2+-sensitive BK and IK potassium channels, and volume-sensitive VRAC chloride channels, in progression of the primary brain cancer glioblastoma. The Authors provide a comprehensive review of the literature, including numerous findings from their own laboratory. The innovative aspect of this review is its focus on the role of hypoxia in modulation of ion channel properties and glioma cell behavior.
This manuscript is expected to generate a significant interest in the field. Minor suggestions for improvement are related to additional citations, clarity, and grammar. It would be helpful to ask for proofreading of this submission by a native English speaker.
Specific suggestions:
[1] In addition to the proposed by the Authors pro-tumorigenic role of Piezo1, activation of this channel also enhances chemosensitivity of GBM cells and is thought to be therapeutically relevant (see and quote S.V. Knobauch et al., S+ACS Omega, 8: 16975-16986, 2023).
[2] The prognostic value of Piezo1 in GBM was recently explored by S. Qu et al. (Cancer Manag Res 12: 3527-3536, 2020).
[3] The role of LRRC8/VRAC channels in glioblastoma proliferation and chemosensitivity has been directly tested in the study by S. Rubino et al. (Front. Oncol., 8: 142, 2018).
[4] There was a negative finding that LRRC8/VRAC channels do not regulate either cell proliferation or cell migration in two established GBM cell lines, U87 and U251 (T. Liu and T. Stauber, Int. J. Mol. Sci., 20: 2663, 2019)). This opposing point of view deserves brief mentioning.
[5] There was a descent review by R. Xu on the role of LRRC8/VRAC channels in diverse cancers (Int. J. Biol. Macromol. 159: 570-576, 2020). Perhaps, it deserves additional mentioning in the present work.
[6] In Fig. 1, there is a typo: Necrotic area has <1% oxygen tension levels (not >1%).
[7] Lns. 235-237. The following two sentences are grammatically problematic: “However, the recent cryo-EM structure of the Piezo1 clearly shows that the channel is a trimer [105]. By contrast, Piezo1 have selective activators as yoda1 and jedi2, [106,107].”
[8] Lns. 461-462. Typo: “…they may be modulated by the HYPOXIA (or HYPOXIC insult)”.
[9] It seems that many publications in the reference list are missing page range information and/or article numbers (e.g., [10, 20, 28, 36, 39, 49, 63, 66, 80, 92-95, 106, 116, 118].
Comments on the Quality of English Language
Additional language proofreading would be helpful.
